# Hippocampal Neurogenesis in Alzheimer’s Disease: Multimodal Therapeutics and the Neurogenic Impairment Index Framework

**DOI:** 10.3390/ijms26136105

**Published:** 2025-06-25

**Authors:** Li Ma, Qian Wei, Ming Jiang, Yanyan Wu, Xia Liu, Qinghu Yang, Zhantao Bai, Liang Yang

**Affiliations:** Research Center for Natural Peptide Drugs, Shaanxi Engineering & Technological Research Centre for Conservation & Utilization of Regional Biological Resources, Yanan University, Yanan 716000, China; mlx010305@163.com (L.M.); weiqian@yau.edu.cn (Q.W.); jm@yau.edu.cn (M.J.); wyy@yau.edu.cn (Y.W.); xliu@yau.edu.cn (X.L.); qhyang@yau.edu.cn (Q.Y.)

**Keywords:** Alzheimer’s disease, Neurogenesis Impairment Index, neurotrophic factor, natural products, neurotransmitter

## Abstract

Alzheimer’s disease (AD) is characterized by progressive cognitive decline strongly associated with impaired adult hippocampal neurogenesis (AHN). Mounting evidence suggests that this impairment results from both the intrinsic dysfunction of neural stem cells (NSCs)—such as transcriptional alterations in quiescent states—and extrinsic niche disruptions, including the dysregulation of the Reelin signaling pathway and heightened neuroinflammation. Notably, AHN deficits may precede classical amyloid-β and Tau pathology, supporting their potential as early biomarkers of disease progression. In this review, we synthesize recent advances in therapeutic strategies aimed at restoring AHN, encompassing pharmacological agents, natural products, and non-pharmacological interventions such as environmental enrichment and dietary modulation. Emerging approaches—including BDNF-targeted nanocarriers, NSC-derived extracellular vesicles, and multimodal lifestyle interventions—highlight the translational promise of enhancing neurogenesis in models of familial AD. We further propose the Neurogenesis Impairment Index (NII)—a novel composite metric that quantifies hippocampal neurogenic capacity relative to amyloid burden, while adjusting for demographic and cognitive variables. By integrating neurogenic potential, cognitive performance, and pathological load, NII provides a framework for stratifying disease severity and guiding personalized therapeutic approaches. Despite ongoing challenges—such as interspecies differences in neurogenesis rates and the limitations of stem cell-based therapies—this integrative perspective offers a promising avenue to bridge mechanistic insights with clinical innovation in the development of next-generation AD treatments.

## 1. Introduction

Alzheimer’s disease (AD) is a progressive neurodegenerative disorder characterized by a gradual onset and eventual manifestation of dementia symptoms. Clinically, it manifests as cognitive decline—including memory loss, spatial disorientation, executive dysfunction—as well as psychological and behavioral disturbances [1]. Adult hippocampal neurogenesis (AHN), defined as the continuous generation and functional integration of new neurons in the dentate gyrus, plays a critical role in maintaining cognitive plasticity, particularly in learning and emotional memory consolidation [2,3]. In AD, a marked decline in neurogenic capacity has been observed, likely attributable to impaired proliferation, differentiation, and survival of neural stem and progenitor cells [4,5]. Given the functional importance of AHN, multiple research teams have investigated its impairment in transgenic models of familial AD. AHN contributes to diverse cognitive domains, including cognitive flexibility—partly mediated by brain-derived neurotrophic factor/tropomyosin receptor kinase B (BDNF/TrkB) signaling that regulates dendritic complexity of newborn neurons [6]—as well as emotional memory, spatial navigation, novelty detection, and pattern separation [7]. This reduction in AHN is thought to contribute to cognitive deficits observed in AD patients. While a correlation between AHN impairment and AD progression has been established, the precise temporal relationship and causal interplay between AHN dysfunction and classical AD pathological hallmarks (e.g., Aβ plaques, Tau tangles) remains subjects of ongoing debate. Recent findings suggest that AHN disruption may precede the accumulation of neuropathological features. In transgenic mouse models such as *APP*swe/*PS1*ΔE9, *5×FAD*, and *3×Tg*-AD, significant reductions in neural progenitor cell proliferation and neurogenesis have been observed as early as 2–3 months of age—prior to the detectable deposition of Aβ plaques or Tau hyperphosphorylation [8,9,10,11]. Evidence from human postmortem studies further supports this temporal sequence. Moreno-Jiménez et al. reported a progressive decline in doublecortin-positive (DCX+) immature neurons in the hippocampus across Braak stages in AD patients [3]. Similarly, Tobin et al. found that individuals with mild cognitive impairment (MCI) already exhibit significant AHN deficits, suggesting early neurogenic involvement prior to clinical dementia [12]. Moreover, single-nucleus RNA sequencing (snRNA-seq) of the dentate gyrus(DG) in AD patients has revealed a loss of neuronal subtypes with immature transcriptional profiles, along with downregulation of genes related to synaptic plasticity [13,14]. In vitro studies using induced pluripotent stem cells (iPSCs) derived from both familial and sporadic AD patients have also shown early impairments in neurogenic programs, further supporting the notion that AHN disruption may be an early and intrinsic feature of AD pathogenesis [14,15,16,17].

In normal adult mammals, neural stem cells (NSCs) in the subgranular zone (SGZ) of the hippocampal DG undergo a tightly regulated sequence of self-renewal, proliferation, migration, and differentiation. These NSCs give rise to new granule cells that migrate into the granule cell layer (GCL), extend dendritic processes into the molecular layer, and integrate into pre-existing hippocampal circuits [18,19,20]. Throughout the lifespan, the DG maintains a dynamic capacity for neurogenesis and neuronal integration, participating in multisensory memory encoding, enhancement of synaptic plasticity, and improvement learning and memory functions [21]. However, in AD, this neurogenic process is profoundly impaired due to both intrinsic NSC dysfunction and extrinsic niche alterations [2]. Recent studies have identified two distinct subtypes of NSCs, termed α-NSCs and β-NSCs, which exhibit differential responses to environmental cues. Single cell transcriptome profiling has revealed different molecular signatures between these subtypes, highlighting protein arginine methyltransferase 1 (PRMT1) as a key regulatory element. Inhibiting of PRMT1 specifically enhances the neurogenic ability of β-NSCs (2-fold vs. α-NSCs) and promotes cognitive function in aged and AD mouse models [22]. This dual disruption—comprising both cell intrinsic NSC deficits and extrinsic niche degradation—results in reduced hippocampal neurogenesis, which has been strongly associated with the severity of cognitive dysfunction in AD patients.

Emerging evidence delineates three major pathological intersections between AHN and AD progression. **(1) Neurogenic niche dysfunction**: Zhao et al. demonstrated key mechanisms underlying adult neurogenesis and its potential involvement in neurodegenerative diseases [23]. Similarly, Winner et al. reviewed evidence suggesting that enhancing AHN could mitigate cognitive decline in neurodegenerative disorders such as AD [24]. In addition, signaling pathway dysregulation has been implicated in AHN-AD crosstalk. Specifically, the Reelin signaling pathway—which regulates neurodevelopment, neurogenesis, and synaptic plasticity—interacts with apolipoprotein E4, Aβ, and Tau, all of which are central to AD pathogenesis [25]. In AD, elevated hippocampal iron level and increased expression of inflammatory mediators, such as tumor necrosis factor (TNF), further impair neurogenesis [26]. These findings demonstrate that augmenting neurogenesis may ameliorate AD-associated memory loss and emotional disturbances.

**(2) Temporal precedence of AHN decline**: Salta et al. reported that reductions in AHN markers can occur before the onset of significant cognitive decline and, in some cases, may precede the detectable accumulation of Aβ plaques or phosphorylated Tau (pTau), particularly in early disease stages and animal models [2,12,15,27]. Neurogenesis impairment has been observed in *APP/PS1* mice at 6 months of age, in *5×Tg*-AD (*APP KM670/671NL* (Swedish), *APP I716V, APP V717I, PSEN1 M146L* (A>C), *PSEN1 L286V*) mice before 2 months of age, and in *3×Tg*-AD (*APP KM670/671NL* (Swedish), *MAPT P301L, PSEN1 M146V*) mice at 4 months of age [28]. Tobin et al. further confirmed that the number of immature neurons in the hippocampus correlates with cognitive decline in AD patients [12]. Colussi et al. reported a ~72.7% reduction in DCX+/BrdU+ cells in the hippocampus of AD model mice. While the study utilized a conditional knockout of Nup153 in hippocampal NSC, the broader expression of Nup153 across the brain highlights the need for cell-specific targeting in translational application [29]. Collectively, these findings suggest that AHN dysfunction arises early in AD and may contribute to subsequent pathological and cognitive outcomes [12].

**(3) Circuit-level integration failures**: Comparative studies in AD mouse models (*5×FAD and 3×Tg-AD*) reveal that increased adult neuronal populations in the supramammillary nucleus are associated with molecular changes in hippocampal CA1 and CA3 regions, including elevated pCaMKII/pCREB expression (50% increase, *p* < 0.05) and altered postsynaptic potentials [30]. Zhang et al. reported that newborn neurons in APP/PS1 mice exhibit morphological and functional integration defects, which were reversed by inhibiting aberrant AHN, thereby improving cognition [31]. Additionally, Moreno-Jiménez et al. demonstrated that AHN is compromised in both human and animal models during aging and neurodegeneration, impairing memory formation and retrieval [3]. AHN dysfunction in AD is closely associated with the activation state of CaMKII/CREB signal, indicating that hippocampal hyperexcitability underlies neurogenic impairment [32].

Zheng et al. reported that the accumulation of pTau in AD leads to functional impairment of GABAergic interneurons in the dentate gyrus, characterized by reduced inhibitory synaptic transmission. This disruption contributes to NII. Restoration of GABAergic neurons, or pharmacological enhancement of tonic inhibition via δ-subunit-containing GABA_A receptors (δ-GABA_ARs) using 4,5,6,7-tetrahydroisoxazolo(5,4-c)pyridin-3-ol (THIP, was shown to restore NII and ameliorate cognitive deficits in both AD patients and animal models [33,34]. Additionally, GABAergic activation suppresses neuroinflammation through TLR4/MyD88 signaling [35]. A single systemic injection of wild-type hematopoietic stem/progenitor cells in *5×FAD* mice resulted in preserved cognitive performance in novel object recognition tasks. The treatment also partially maintained blood–brain barrier integrity and reduced amyloid plaque accumulation, microgliosis, and neuroinflammation [36]. Apodaca et al. demonstrated that extracellular vesicles derived from human NSCs can protect against behavioral and molecular pathology amylosis and neuroinflammation by significantly inhibiting Aβ plaques aggregation (the number of dense core Aβ plaques in the detection area of late-stage AD mice treated with EV decreased by about 20–30%) and microglia activation [37]. This aligns with emerging concepts of systemic regulation of neurogenesis via blood-borne factors [38].

Immature dentate granule cells (imGC), generated by AHN, contribute to brain plasticity and are dysregulated in various human neurological diseases [39]. Zhou et al. determined that imGCs persist in the human hippocampus across the lifespan, with age-related decline in their proportion [13]. However, whether imGC loss is a cause or consequence of amyloid pathology remains debated. Compared to mature granulosa cells (mGCs), imGCs exhibit distinct neurogenesis- and plasticity-related gene expression, which is altered in AD [13]. Understanding the impact of AHN on AD pathology is critical for developing strategies to preserve cognitive function and neuronal integrity in affected individuals.

To date, no study has concurrently evaluated (a) the temporal window of AHN therapeutic susceptibility relative to AD progression, and (b) the comparative efficacy of intrinsic neurogenic modulation versus extrinsic niche engineering. Despite advances in AHN-targeted therapies, there remains a lack of systematic evaluation of pharmacological versus non-pharmacological interventions, especially regarding their long-term and combinatorial effects. Here, we centered on neurotrophic factors, neurotransmitters, natural compounds, physical activity, and enriched environments in regulating AHN within AD contexts. These insights highlight the significance of elucidating the complex interplay between AHN and AD and support the development of multimodal interventions. We hypothesize that combinatorial AHN modulation may synergistically alleviate AD-related cognitive deficits by targeting both neurogenic niche and systemic homeostasis.

To this end, we propose the NII—a novel, multidimensional metric designed to quantify the neurogenic potential relative to pathological burden, while accounting for individual variability in age, sex, and cognitive status. The NII integrates key modulators of adult neurogenesis, largely derived from preclinical studies in mouse models of AD, and aims to support the development of individualized neurogenic interventions. This index encompasses three core domains: (1) Demographic variables: age-stratified in 6-month intervals and sex-specific factors (e.g., estrous cycle in rodent models), reflecting modulatory roles in neurogenesis and pathology. (2) Cognitive metrics: Cognitive status is assessed using Montreal Cognitive Assessment (MoCA) scores [40], with z-scores ≤ 2 standard deviations from age-matched norms, and in an AD model, mice cognitive status is assessed using rodent-specific cognitive tests (e.g., Novel Object Recognition Test, Morris Water Maze), with performance quantified using z-scores calculated relative to age-matched control groups (z ≤ −2 indicating impairment). Where available, cerebrospinal fluid (CSF) biomarker ratios (e.g., Aβ42/pTau) are integrated as quantitative for pathological burden and progression. (3) Neurogenic capacity [41,42]: calculated using the following formula: Neurogenic Capacity = (DCX + cells/SGZ volume) × (PSA-NCAM density/Aβ plaques density). This composite reflects both structural and pathological modulators of neurogenesis, as demonstrated in animal models. However, due to ethical and technical limitations, such direct quantification is not feasible in living human subjects. Translation to clinical use will require the identification and validation of surrogate markers—such as PET imaging ligands for neurogenesis, peripheral blood correlates, or CSF indicators of neural progenitor activity.

Taken together, the NII offers a scalable framework for quantifying neurogenic integrity, with each dimension tailored to the available modalities and ethical feasibility of the species under investigation. While full implementation currently remains restricted to preclinical research, this construct may serve as a foundation for translational alignment between basic and clinical neurogenesis studies.

Annotations: DCX cells, which are widely accepted markers of newly generated granule cells [43]; polysialylated form of neural cell adhesion molecule (PSA-NCAM), a neuroplasticity marker, which may prove to be a potential target to facilitate repair/regeneration after CNS injury [44]; SGZ, where NSCs of the hippocampal DG undergo self-renewal, proliferation, migration, and differentiation [45]; amyloid-β (Aβ) plaques, which are the main characteristic of AD pathology in the brain, and their density can characterize the pathological process of AD [46].

## 2. Hippocampal Neural Stem Cells and AD Pathologic Processes

In AD, pathology begins in the transentorhinal cortex and locus coeruleus and progressively affects the hippocampus, resulting in its hallmark atrophy [47,48,49]. Consequently, the reconstruction or repair of hippocampal nerves is considered an effective strategy for alleviating AD symptoms. In AD patients, the hippocampal region exhibits an abnormal proliferation of NSCs and disrupted neurotrophic signaling. Increased NSC quiescence results in a greater tendency to differentiate into non-neuronal subtypes. Furthermore, excessive Aβ precursor protein promotes glial differentiation, depleting the NSC pool and potentially accelerating neurodegeneration [50,51] (as summarized in Figure 1). This pathological shift fundamentally disrupts the neurogenic niche, creating a self-reinforcing cycle of cognitive decline.

This tightly regulated equilibrium highlights the vulnerability of the neurogenic niche to AD-related insults. In the healthy DG, NSCs in the SGZ undergo self-renewal, proliferation, migration, and differentiation. These cells migrate into the GCL, differentiate into granule cells, and extend into the molecular layer, integrating into the existing hippocampal circuits [13,52]. Adult DG neurogenesis is continuous throughout life and contributes to multisensory memory encoding, synaptic plasticity, and cognitive function [3,4]. Understanding these baseline processes is essential for appreciating how AD disrupts a finely tuned regenerative system.

To counteract AD-driven neurogenic collapse, emerging therapeutic strategies have focused on two complementary approaches: endogenous NSC activation and exogenous NSC transplantation. Enhancing the survival, proliferation, and differentiation of transplanted or endogenously activated NSCs is critical for therapeutic efficacy. Addressing the underlying causes of NSC quiescence and misdifferentiation in AD may lead to improved interventions. Both the induction of endogenous NSC proliferation and exogenous NSC transplantation have shown promise in partially restoring AD pathology. Reprogramming glial cells into multipotent neural progenitors can also generate new neurons and repair brain injuries, as demonstrated in models of traumatic brain injury [53,54]. Transplanting NSCs into early-stage AD mouse model repairs damaged dendritic spine and enhances learning and memory [55,56]. Similarly, stem cell transplantation into AD rat models significantly reduces Tau phosphorylation and Aβ42 plaque load, while also enhancing anti-inflammatory and anti-apoptotic effects, resulting in improved spatial memory and cognitive outcomes [57]. These studies collectively underscore the critical role of NSC development in maintaining physiological homeostasis in AD.

Inducing glial cell reprogramming or neural stem cell transplantation to address AD pathology faces significant drawbacks, such as immune response, low survival rate, poor proliferation and migration ability, and limited directed differentiation ability. While these therapeutic approaches show promise, significant challenges hinder their clinical translation. Therefore, understanding the growth patterns of NSCs and screening novel induction methods that promote directed proliferation, differentiation, and migration are effective strategies for regulating NSCs for AD treatment.

Despite their therapeutic potential, strategies such as glial reprogramming and NSC transplantation face significant limitations, including immune rejection, low survival rates, poor migration capacity, and limited directed differentiation. These challenges currently hinder clinical translation. Therefore, understanding NSC growth dynamics and identifying novel induction protocols that enhance proliferation, lineage-specific differentiation, and targeted migration represent key steps in advancing NSC-based therapies for AD. In the study of AD pathogenesis and treatment development, it is imperative to integrate the concept of neurogenic homeostasis disruption with established hypotheses involving Aβ accumulation, Tau hyperphosphorylation, neuroinflammation, and excitotoxicity. To facilitate this integration, we propose the NII as a novel, multidimensional framework that quantifies neurogenic deficits in relation to pathological burden and cognitive decline. At present, the NII is primarily intended for preclinical applications, particularly as a readout in neurogenesis-targeted treatment trials using AD animal models. It may also serve as a bridge between mechanistic findings and personalized therapeutic strategies. Notably, the rate of adult hippocampal neurogenesis in human (0.004% new neurons per day) is approximately five times lower than that in rodents (0.02%), which necessitates prolonged therapeutic windows—8 to 12 months in humans versus 2 to 4 weeks in mice [58]. These interspecies differences emphasize the need for translational frameworks such as the NII to guide both experimental interpretation and therapeutic timing. Based on these insights, future directions in AD research and intervention may include (1) developing neurogenesis-based indices (e.g., the NII) as a quantitative biomarker for pathological staging and diagnostic evaluation; (2) designing neurogenesis-modulating strategies to enhance cognitive performance and emotional regulation in both preclinical and clinical settings; (3) prioritizing drug discovery targeting key neurogenic regulators, such as BDNF/TrkB signaling, the Wnt/β-catenin pathway, and Notch1 receptor activity. These pathological alterations underscore the urgency of exploring therapeutic strategies that address both intrinsic NSC dysfunction and extrinsic niche regulation. The translational potential of these findings warrants a systematic evaluation of neurogenic modulation across AD progression stages.

## 3. Research on Adult Neurogenesis in Alzheimer’s Disease

The progressive atrophy of the cerebral cortex and neurodegenerative changes are the pathological basis for the progressive worsening of cognitive dysfunction in patients with AD [59]. During the course of AD, the entorhinal cortex and hippocampus are the first regions affected, followed by the spread of the disease, which eventually involves the entire cerebral cortex [60]. A series of structural and morphological changes occur in the hippocampus and cortex during AD pathogenesis. As early as 1982, a U.S. team first demonstrated that the loss of specific neuronal populations contributes to subcortical damage in the brains of AD patients [61]. In 2019, Spanish scientists discovered that neurogenesis persists in the brains of individuals aged 43 to 87 years. However, notably, neurogenesis is rarely observed in the brains of AD patients, indicating a close association between memory impairment and disrupted hippocampal neurogenesis [3].

In 2022, Zhou et al. utilized snRNA-seq to show that imGCs persist in the hippocampus across the lifespan. In infancy, imGCs account for 9.4% of total granule cells, declining to 3.1–7.5% after age 4 and gradually decreasing with age. Compared to mGCs, imGCs display gene expression profiles associated with neurogenesis and synaptic plasticity. In AD patients, the proportion of imGCs is significantly lower than age-matched controls, accompanied by the downregulation of genes related to synaptic plasticity and signal transduction [13].

Given the impaired neurogenesis in AD, exogenous interventions such as NSC transplantation have been explored to compensate for neuronal loss. For example, transplanted human NSCs (huCNS-SC) into the lateral ventricles of transgenic AD model mice demonstrated effective regional integration, with the transplanted cells exhibiting migration and differentiation capabilities. Following transplantation, reductions were observed in Tau phosphorylation and Aβ production, while the deactivation of microglial cells led to reduced neuroinflammation and apoptotic signaling. These molecular improvements were accompanied by enhanced synaptic plasticity. Behavioral tests further indicated improved spatial memory in AD mice receiving NSC transplantation, compared to controls [62,63].

These findings collectively suggest that both AD patients and animal models experience a disruption of hippocampal NSC developmental homeostasis, exacerbating neurodegeneration. NSC transplantation has been shown to reduce inflammation and improve memory performance in AD animal models. Moreover, recent studies have employed stem cells derived from the placenta, adipose tissue, umbilical cord blood, and bone marrow to the research and treatment of AD [64,65,66,67,68]. Nevertheless, NSC transplantation remains at an early experimental stage. Therefore, promoting both endogenous and exogenous hippocampal neurogenesis under pathological conditions will undoubtedly become a new direction in the exploration of AD diagnosis and treatment strategies.

## 4. Neurotransmitter and Signaling Molecule Modulation of Hippocampal Neurogenesis Alleviates Alzheimer’s Disease Pathology

Neurotransmitters, along with certain adenosine analogs, play crucial roles in neurogenesis. The primary neurotransmitters involved are acetylcholine (ACh), γ-aminobutyric acid (GABA), and the signaling molecule cyclic adenosine monophosphate (cAMP). Cholinergic neurons are specialized for the release of ACh and contain the enzymes choline acetyltransferase (ChAT) and acetylcholinesterase (AChE) within their cytosol. These neurons include local interneurons and long-range projection neurons and are widely distributed throughout the central nervous system, where they can interact with non-cholinergic neuronal elements. Research on aging and neurodegeneration has increased its focus on the cholinergic system, particularly in the context of AD [69].

According to Braak and Braak, the earliest pathological changes in AD occur in the locus coeruleus and transentorhinal cortex, preceding the degeneration of cholinergic neurons in the basal forebrain and their cortical projections. This degeneration contributes to cognitive decline in later stages. Jing et al. reported that the activity of ChAT in the basal forebrain of AD mice decreased at 6 months of age, with cholinergic neuron loss observed by 8 months. Concurrently, choline levels in the basal forebrain, the expression of N-acetyl aspartate in the hippocampus, basal forebrain ChAT and vesicular acetylcholine transporter (vAchT), and the expression of muscarinic acetylcholine receptor2 (CHRM2) in the hippocampus were all found to be increased [70]. In addition, Lancao et al. demonstrated the activation of AMPAR and increased ACh levels, which enhanced neurogenesis, neuronal differentiation, and hippocampal activity [71].

GABA serves as a pivotal inhibitory neurotransmitter in the mammalian central nervous system. In the adult mouse hippocampus, neural precursor cells arise from a subset of interneurons that express albumin and receive GABAergic synaptic inputs. The cognitive decline observed in AD is significantly influenced by the accumulation of pTau [72], which impairs AHN. The intracellular aggregation of Tau within GABAergic interneurons leads to circuit imbalances and neurogenic dysfunction. Proteomic and phosphoproteomic analyses revealed that under the physiological conditions of AD, the ventral hippocampal CA1 (vCA1) region gradually accumulates Tau. Notably, the hyperexcitability of parvalbumin (PV) interneurons in this region is associated with the intracellular accumulation of mislocalized, hyperphosphorylated Tau, impairing neuronal excitability and disrupting circuit function. The overexpression and accumulation of human Tau in PV neurons significantly inhibits excitability and suppresses the clear discriminative firing in vCA1 [73]. These findings suggest that enhancing GABAergic neurotransmission may be a promising therapeutic strategy for targeting neuronal precursors or affected cells in AD [33]. Moreover, the chemical genetic inhibition of excitatory neurons or pharmacological enhancement of GABA can rescue Tau-induced AHN deficiency and improve situational cognition [33]. Bao et al. demonstrated that GABAergic neural networks regulate the quiescence of NSCs and hippocampal neurogenesis via dynamic interactions between long-range GABA-projecting and local PV interneurons [74].

The cAMP-responsive element-binding protein (CREB) signaling pathway plays a fundamental regulatory mechanism for adult neurogenesis, encompassing the survival, maturation, and integration of newly generated neurons into existing networks [75,76]. Ginsenoside RK3 promotes neurogenesis and synaptic development through CREB/BDNF signaling, enhancing learning and cognition in APP/PS1 and C57 mice [75]. CREB signaling is implicated in various cognitive and neurodegenerative disorders. Notably, NO/cGMP/CREB signaling is downregulated in aging and neurodegenerative diseases and is disrupted by Aβ and Tau pathology. Under physiological conditions, the NO/cGMP/PKG/CREB pathway interacts with Aβ to support long-term potentiation and memory consolidation [77,78,79].

The AchE inhibitor donepezil exerts neuroprotective effects by upregulating CREB (via MT/ERK/CREB signaling) phosphorylation levels in the hippocampus, thereby mitigating memory deficits triggered by transient cerebral ischemia [80]. Furthermore, CREB is also hypothesized to participate in learning and memory processes via its regulation of adult neurogenesis [81]. In AD, the dysfunction of PV interneurons contributes to cortical hyperexcitability, manifesting as subclinical epilepsy and abnormal gamma oscillations. Spoleti et al. used a Tg2576 AD mouse model to demonstrate that reduced hippocampal dopaminergic innervation caused by VTA dopaminergic neuron degeneration impairs PV neuron firing and gamma wave activity, leading to hippocampal overexcitation via reduced D2 receptor-mediated CREB activation. Treatment with L-DOPA or the selective D2 receptor agonist quinpirone rescues p-CREB levels, improves PV IN mediated inhibition, and reduces overexcitement [82].

Tau pathology also suppresses key neurogenic signaling pathways, including PKA/CREB/BDNF/TrkB and PKA/GluA1 signaling cascades, thereby contributing to synaptic and memory impairment [83]. Yan et al. found that curcumin alleviates spatial memory impairment by reversing Tau abnormalities and reducing p-CREB in the hippocampus [84]. Similarly, Ali et al. demonstrated that melatonin alleviates Aβ 1-41-induced neurotoxicity by restoring p-CREB (Ser133) and downstream synaptic proteins, while also reducing Tau hyperphosphorylation and neurodegeneration via PI3K/Akt/GSK3β signaling [85]. A summary of the roles of neurotransmitters and signaling molecules in neurogenesis and AD pathology is presented in Table 1.

## 5. Neurotrophic Factors Promote Hippocampal Neurogenesis to Ameliorate AD Pathology

Beyond metabolic substrates, neurotrophic factors (NTFs) serve as key molecular regulators of hippocampal neurogenesis, directly influencing AD pathology. Neurogenesis requires substantial nutritional support, including glucose, amino acids, and lipids. It is also influenced by the surrounding environment, such as the extracellular matrix and intercellular signaling molecules. Together, these factors regulate neurogenesis, ensuring proper neuronal development and function. Abnormalities in this process can lead to neurodevelopmental and other neurological disorders.

Major NTFs—including BDNF, nerve growth factor (NGF), neurotrophin-3 (NT-3), and neurotrophin-4 (NT-4)—promote neuronal proliferation, differentiation, injury repair, and survival [86]. Additionally, NTFs are involved in long-term potentiation and inhibition, regulating synaptic transmission and plasticity. The CA1 and CA3 regions of the hippocampus, along with the DG, are key sites for BDNF synthesis. Increasing BDNF expression promotes neural stem cell growth and can ameliorate mood disorders [9,87]. Both BDNF and NGF are known to enhance neural development and repair CNS lesions, as well as cognitive deficits associated with AD [88]. In AD, TrkB-FL cleavage triggered by Aβ impairs BDNF signaling, thereby impairing neuronal survival, differentiation, synaptic transmission, and plasticity. João Fonseca Gomes et al. found that TrkB-FL cleavage occurred in the early stages of the disease and increased with the severity of pathology in cerebrospinal fluid and post-mortem brain samples. They designed a small TAT fusion peptide, in which the TAT-TrkB peptide with a lysine–lysine linker was able to prevent TrkB-FL cleavage in vitro and in vivo and rescued synaptic defects induced by oligomer Aβ in hippocampal slices. Additionally, it was found that in the *5×FAD* AD mouse, this TAT-TrkB peptide improved cognitive ability, improved synaptic plasticity defects, and prevented Tau pathological progression in vivo [89].

Neurotrophic factors and NSCs have emerged as prominent areas of research for AD therapy [90,91]. NTFs are essential proteins for the survival, development, and functional maintenance of nerve cells, promoting cell growth and maintenance in the nervous system. Neural stem cells, with their potential to self-renew and differentiate into multiple neural cell types, are crucial for the repair and regeneration of the nervous system. Research indicates that modulating neurotrophic factor levels can enhance neuron survival and function, potentially alleviating symptoms in AD [92,93]. Neural stem cell therapies are also promising, as these cells can differentiate into various damaged nerve cell types and help repair affected neural networks [94]. The elevated expression of the myelin formation-related gene Lingo1 significantly activates the RhoA/ROCK1 signaling pathway through its interaction with NgR and p75NTR, subsequently promoting myelin loss and the abnormal phosphorylation of Tau protein. The upregulation of Lingo1 in hippocampal neurons further inhibits the EGFR/PI3K/Akt pathway, which may increase neuronal apoptosis [95]. Lin Shen et al. demonstrated that the genetic reduction of p75NTR in P301L mice rescued memory deficits, alleviated Tau protein hyperphosphorylation, and restored the activity of the AKT/GSK3β pathway. They support the crucial role of p75NTR in Tau-paGSK pathology as a potential drug target for frontotemporal degenerative Tau and related Tau diseases [96].

The peripheral blood BDNF levels in AD patients were significantly lower than those in healthy controls (24 studies, Hedges’ g = −0.339, 95% confidence interval (CI) = −0.572 to −0.106, *p* = 0.004) [97]. Serum BDNF levels were associated with the rate of cognitive decline in AD, and serum proBDNF levels were associated with hippocampal proBDNF levels, which were associated with hippocampal pTau expression [98]. BDNF gene delivery or exogenous supplementation can reverse cognitive deficits in AD model animals. Multiple studies support the feasibility of improving cognitive function in AD through the exogenous supplementation of BDNF. For example, the use of biodegradable nanocarriers (exosomes) combined with intranasal administration technology can avoid invasive procedures and enhance the brain-targeted distribution of BDNF [68,91]. Rosenberg et al. demonstrated that NGF gene therapy may enhance cholinergic neuron survival by reversing neuronal death or atrophy. Rats receiving NGF-secreting grafts exhibited a 92% survival rate, with 49% of cholinergic cells preserved. Chen et al. further investigated the effects of nerve growth factor (NGF) on rodent behavior. They bilaterally injected NGF-producing fibroblasts or non-infected fibroblasts into the nucleus basalis (a brain region undergoing degeneration in AD) of behaviorally impaired aged rats and cognitively intact young adult rats. Behavioral tests revealed significant cognitive improvement in the NGF-treated aged group. Immunohistochemical staining confirmed an increased number of NGF receptor-positive neurons in the graft-treated aged rats [99].

These are encouraging findings, but there are still challenges in the clinical application of NTFs and NSCs in AD treatment. The key challenges include ensuring the safety and effectiveness of treatment, as well as developing effective delivery methods. In the context of AD monitoring and therapeutic biomarker development, the incorporation of neurotrophic factors (NTFs, e.g., serum BDNF levels) into a dynamic monitoring framework incorporating serum biomarkers and tissue expression profiles is warranted. This system should emphasize their dual-pathway modulation of Aβ and Tau pathology while identifying synergistic regulatory mechanisms and potential therapeutic targets. However, caution should be exercised regarding dosage optimization in therapeutic strategies to prevent aberrant neurogenic overactivation, which may exacerbate circuit hyperexcitability. In translational research, particular attention should be paid to interspecies variations in neurogenic rates and trophic factor expression patterns between humans and AD animal models. A summary of the effects of neurotrophic factors in neurogenesis and AD pathology is tabulated in Table 2.

## 6. Natural Product Active Ingredients Promoting Neurogenesis to Improve AD Pathology

Numerous studies have demonstrated that natural compounds can stimulate the proliferation and differentiation of neural stem cells, thereby enhancing learning and memory functions in models of familial AD. For instance, resveratrol has been shown to modulate SIRT1, promoting adult hippocampal neurogenesis. This compound consistently enhances neurogenesis and improves cognitive functions in AD animal models through Wnt/β-catenin signaling via the direct phosphorylation of GSK-3β at Ser9 (inhibitory site), thereby stabilizing β-catenin to induce NeuroD1-mediated neuronal differentiation [100,101]. Similarly, curcumin upregulates the expression of genes such as Notch1 and Hes1, along with CDK4 and cyclin D1. It also increases the level of NICD and Hes1 proteins, thereby promoting neurogenesis in APP/PS1 mice and significantly improving their learning and memory capabilities [102]. Furthermore, curcumin enhances adult neurogenesis in AD mice through targeted pathways including PI3K/AKT, GSK3β/Wnt/β-catenin, and CREB/BDNF [103]. Coronarin D has been shown to significantly promote the differentiation of NSCs into astrocytes via the JAK/STAT signaling pathway, leading to an xincrease the expression of GFAP at both mRNA and protein levels [104]. Additionally, the flavonoid trichostatin enhances neuronal differentiation of NSCs and facilitates axon growth. At 100 μM (*n* = 5, *p* < 0.05), Troxerutin completely reversed Aβ42-induced axonal growth inhibition and promoted significant cell migration [105].

Moreover, flavonoids derived from Scutellaria baicalensis stems and leaves have been reported to ameliorate Aβ-induced learning and memory deficits through the CaM-CamkIV-CREB signaling pathway, thereby regulating neuroplasticity [106]. Farnesol 3,6′-disinapoyl sucrose (DISS, active constituent of Polygala tenuifolia) significantly enhances neural stem cell proliferation and neuronal differentiation in APP/PS1 mice. Treatment with DISS (20 mg/kg) for four weeks effectively rescued cognitive deficits, neuronal damage, and impaired neurogenesis in adult transgenic mice [107]. Similarly, ginsenoside compound CK (5 mg–15 mg/kg) has increased hippocampal neurogenesis in young (2 months) and elderly (24 months) mice (*n* = 12 in 2-mo; *n* = 11 in 24-mo mice, respectively, *p* < 0.01–0.001 vs. vehicle-treated groups) [108]. In our preliminary screening study, the dragon’s blood extract and its active ingredient QLX-N demonstrated the ability to enhance BrdU^+^/Nestin^+^ cell proliferation (116% increase vs. control, *p* < 0.01) and DCX+ neuronal differentiation (2.3-fold elevation, *p* < 0.001) in SGZ, significantly enhancing spatial learning memory and motor abilities in rat models [109,110]. This effect is associated with the induction of neurotrophic factors such as BDNF and the activation of the MAPK/AKT signaling pathway, underscoring their potential to promote endogenous neural stem cell growth [109,110,111,112]. Notably, several natural bioactive compounds have been shown to enhance endogenous neurogenesis. While there is currently no direct evidence linking these compounds to excessive neurogenic outcomes such as epilepsy or carcinogenesis, their long-term safety remains to be fully evaluated. Further preclinical and epidemiological studies are needed to determine whether such neurogenic modulation operates through tightly regulated mechanisms.

Notably, curcumin’s poor oral bioavailability (~1% in rodents) necessitates nano-encapsulation strategies for clinical translation. The biocompatible γ-cyclodextrin metal–organic frameworks (γ-CD MOFs) significantly enhance the apparent solubility and bioavailability of encapsulated bioactive compounds, including curcumin, enabling potential sustained delivery [113]. In vitro digestion studies revealed that curcumin’s apparent solubility increased by 8-fold with bile salts and remarkably by 53-fold when combined with both bile salts and γ-CD MOFs, with corresponding bioavailabilities reaching 2% and 16%, respectively [114]. Additionally, beyond the compounds described above, several other compounds have been extensively investigated. For instance, lactoferrin has been shown to enhance hippocampal neurogenesis and promote astrocyte reprogramming in neural precursor cells [115,116]. Furthermore, compounds including lactoferrin, catalpol, salidroside, and rosmarinic acid have been shown to regulate neurogenesis and related pathological processes [117,118,119]. A summary of the effects of natural bioactive compounds in neurogenesis and AD pathology is tabulated in Table 3.

## 7. Non-Pharmacological Interventions and Future Directions

Progressive cognitive decline in AD places substantial burdens on patients and their families. Consequently, restoring and enhancing cognitive function are crucial aspects in AD treatment. Non-pharmacological approaches aimed at promoting hippocampal neurogenesis are currently being explored, categorized as follows:

Physical exercise intervention: Physical exercise has shown promise in improving cognitive function, potentially by inducing hippocampal neurogenesis in AD patients. This effect may help mitigate neuroinflammation and enhance cognitive capabilities [9,120]. Compared to sedentary mice, exercised mice exhibit significantly elevated expression levels of BDNF and other related factors (hippocampal BDNF increased 1.5-fold, *p* < 0.001) [9]. Exercise-induced BDNF upregulation, with newly generated hippocampal neurons facilitating the re-establishment of neural networks within the hippocampal region: Studies on running-trained adult mice (*3×Tg-AD* mice) have demonstrated elevated levels of cell proliferation (ki67^+^ cells increased 1.67-fold, *p* < 0.001), migration (DCX^+^ neurons in the DG increased 2.0-fold, *p* < 0.0001), and local circuitry remodeling (the number of dendritic spines increased, *p* < 0.05) [121] compared to sedentary counterparts [122,123]. The positive effects generated after these exercises may be closely related to the activation of the BDNF-TrkB signaling pathway [124,125].

Environmental enrichment strategy (EE): Enriched environmental stimulation has been experimentally validated to promote neural dendritic growth, induce hippocampal neurogenesis in mice, and ameliorate cognitive and emotional impairments in AD animal models. Xiaoqin Zhang et al. found that DCX^+^ (4-fold, *p* < 0.001) in the DG region of 4-month-old mice significantly increased after 6 weeks of rich environmental stimulation compared to control mice. At the same time, there was a significant (1.5-fold, *p* < 0.05) increase in learning and memory levels [126]. EE leads to elevated levels of mature NGF and BDNF in AD, which activate the protein kinases/extracellular signal-regulated kinase (MAPK/ERK) and phosphoinositide-3kinase/protein kinase B (PI3k-AKT) signaling pathways, thereby enhancing hippocampal neurogenesis and synaptic plasticity, among other effects [127]. Certainly, we must be acutely aware that approaches such as physical exercise and EE have limited applicability and scope for human patients, as they are not necessarily suitable for individuals with severe cognitive impairment or compromised motor function.

While endogenous stimulation strategies show promise, their efficacy in late-stage AD may be limited by neuronal loss severity, necessitating exogenous cell replacement approaches. Beyond endogenous stimulation, exogenous cell-based approaches have emerged such as ex vivo stem cell therapy, whereby the ex vivo culture and transplantation of stem cells offer another avenue. The proliferation and induced differentiation of stem cells cultured ex vivo, followed by their transplantation in vivo, have shown potential in promoting neurogenesis and facilitating lesion recovery through synaptic integration, representing an allogeneic NSC transplantation paradigm [36,57].

The DG of the hippocampus is characterized by the continuous regeneration and integration of newborn neurons throughout life, with experience-dependent neurogenesis playing a critical role in hippocampal plasticity. Recent data suggest that hippocampal neurogenesis is substantial in the brain, underscoring its significant role in cognition.

However, abnormalities in neural proliferation and differentiation in the hippocampal region, alterations in key signaling proteins, the increased quiescence of neural stem cells, and a tendency to differentiate into non-neuronal subtypes are prevalent in degenerative diseases such as AD [18,128].

Human DG neurogenesis is estimated at ~700 new neurons per day, resulting in ~35% neuronal turnover over the lifespan, as compared to ~10% in rodents [58,129,130]. This physiology of adult neurogenesis has important implications for the treatment of cognitive disorders [39]. Newborn neurons in the DG are essential for pattern segregation, migrating to the dorsal CA3 to form local neural networks involved in the encoding of multiple sensory memories, thus enhancing synaptic plasticity and improving learning and memory [131]. Memory is key to cognitive resilience, and neurodegenerative damage can compromise this process. Therefore, neurogenesis in the hippocampus is vital for maintaining cognitive function and mitigating stress-related damage. Increasing evidence indicates that pro-inflammatory cytokines and inflammatory molecules released during neuroinflammation negatively modulate hippocampal neurogenesis, particularly in the context of AD. For example, IL-1β and TNF-α signaling through the NF-κB pathway has been shown to impair hippocampal neurogenesis by disrupting neural stem cell function. In addition, neuroinflammation can reduce the release of BDNF from cerebrovascular endothelial cells, which normally supports neuronal survival and differentiation, thereby further compromising the neurogenic niche [132]. It should be noted that cross-species differences are emphasized, and the human neurogenesis rate is only 1/5 of that of mice, so the treatment window needs to be extended to 8–12 months.

Given these species-specific differences and disease-associated alterations, promoting hippocampal neurogenesis in the adult brain has been proposed as a novel strategy for treating AD. Patients with neurodegenerative diseases often experience an imbalance in neural stem cell homeostasis, hindering effective neurogenesis activation and exacerbating pathological deterioration. Research has shown that NSCs can proliferate and differentiate continuously in a healthy adult brain, integrating into the existing neural network to replace apoptotic or damaged neurons. Systematic molecular and cellular analyses of hippocampal neurogenic niches in AD patients have been shown to mediate the understanding of the role of AHN in disease progression and recovery. Such investigations will help identify various cell types and states associated with AD etiology, which may respond differently to therapeutic interventions. Thus, regulating the developmental homeostasis of hippocampal NSCs may contribute to a better understanding of AD pathogenesis and support the development of novel diagnostic and therapeutic strategies. However, several inherent limitations should be acknowledged. First, the therapeutic window for effectively targeting AHN may be narrow, as many patients are diagnosed at relatively advanced disease stages when hippocampal circuitry is already significantly compromised. Second, AD pathology is not confined to the hippocampus; the early involvement of regions such as the entorhinal cortex, locus coeruleus, and neocortex—as well as the possibility of pathology spread that bypasses the hippocampal route—suggests that AHN-focused strategies may not be sufficient alone. Third, the frequent presence of co-pathologies (e.g., TDP-43 inclusions, α-synuclein aggregates, cerebrovascular lesions) may diminish the efficacy of neurogenesis-based interventions. Therefore, AHN-directed approaches should be viewed as part of a broader, multimodal therapeutic framework for AD rather than as a stand-alone solution.

Beyond single-modality interventions, recent studies have highlighted the potential of combining pharmacological and non-pharmacological approaches to enhance hippocampal neurogenesis and cognitive function in AD. For instance, combining physical exercise with dietary interventions—such as the Mediterranean diet, rich in antioxidants and anti-inflammatory compounds—has shown promise in improving cognitive outcomes in AD (adjusted differences: MMSE was +0.62, 95% CI +0.18 to +1.05, *p* = 0.005; CDT was +0.51, 95% CI +0.20 to +0.82, *p* = 0.001) [133]. Additionally, innovative drug delivery systems, such as nanoparticle-based delivery methods (drug delivery nanocarriers), by enhancing the blood–brain barrier penetration efficiency of neurotrophic factors (unprecedented 4-fold drug loading capacity compared to traditional methods), are being explored to enhance the efficacy and targeting of neurogenic factors and stem cell therapies [134,135]. These multimodal strategies leverage synergistic interactions between pharmacological, cellular, and lifestyle interventions, significantly enhancing therapeutic efficacy against AD-associated neurodegeneration (as summarized in Figure 2). A summary of the effects of non-pharmacological interventions in neurogenesis and AD pathology is tabulated in Table 4.

## 8. Conclusions

The strategic rejuvenation of hippocampal neurogenesis—via synergistic pharmacological modulation (BDNF/TrkB activation), natural product-mediated NSC priming (curcumin/DISS), and lifestyle interventions (exercise; Mediterranean diet)—represents a paradigm shift in AD therapeutics, with the NII framework enabling stage-specific personalized therapy.

## Figures and Tables

**Figure 1 ijms-26-06105-f001:**
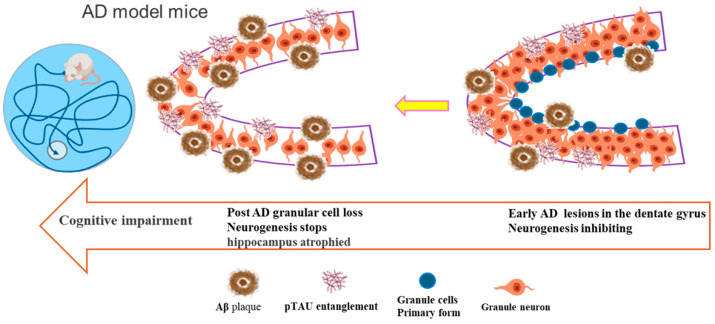
As the pathological progression of AD intensifies, neurogenesis in the hippocampal DG of model mice becomes progressively impaired due to both intrinsic and extrinsic factors, including Aβ and Tau aggregation. Impaired AHN can be an early pathological event in AD progression, occurring ahead of both pronounced amyloid and Tau pathology, as well as cognitive impairment. In AD model animals, the number of newly generated neurons significantly decreases with age, disrupting the stable development of hippocampal NSCs and contributing to progressive impairments in learning and memory abilities.

**Figure 2 ijms-26-06105-f002:**
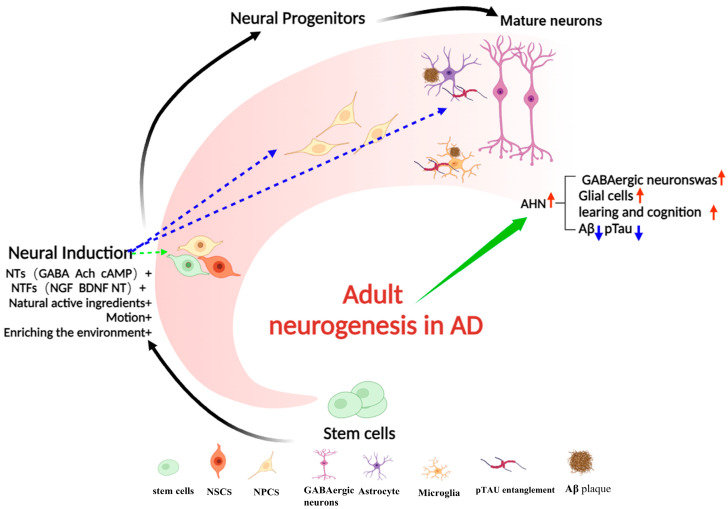
Through regulatory approaches such as neurotransmitters, neurotrophic factors, natural products, exercise and enriched environmental stimulation, and neural stem cell transplantation, hippocampal neurogenesis in AD can be induced. These methods enhance the differentiation activity of neural precursor cells, promote the repair of damaged neural networks, activate glial cells, reduce Aβ and Tau aggregation, and regulate the homeostasis of the microenvironment for neural stem cell development—ultimately improving patients’ learning and memory abilities. Based on this, promoting adult hippocampal neurogenesis may become a novel therapeutic strategy for AD.

**Table 1 ijms-26-06105-t001:** Neurotransmitters/Signaling Molecules in Neurogenesis and AD Pathology.

Molecule	References	Mechanism of Action	Impact on Neurogenesis	Role in AD Pathology
**Acetylcholine** **(ACh)**	[47,48,49]	Released by cholinergic neurons (regulated by ChAT/AchE). Activates AMPAR to enhance ACh levels (Lancao Decoction).	Supports synaptic plasticity and neuronal differentiation. Upregulates hippocampal neural activity.	Early AD pathology involves cholinergic neuron loss in the basal forebrain. Reduced ChAT activity and ACh levels correlate with cognitive decline. Restoring ACh signaling rescues memory deficits.
**GABA**	[17,50,51]	Inhibitory neurotransmitter regulating neural precursor quiescence. Tau accumulation in GABAergic interneurons disrupts GABAergic signaling.	Maintains NSC quiescence via long-range GABAergic inputs. Impaired GABAergic transmission suppresses AHN.	Tau pathology in GABAergic interneurons causes circuit imbalance and AHN deficits. Enhancing GABAergic signaling rescues Tau -induced AHN impairment.
**CREB Pathway**	[52,54,55,58,60,61]	Activated by NO/cGMP/PKG, BDNF/TrkB, or PKA. Regulates neuronal survival, synaptic plasticity, and memory.	Promotes neurogenesis and synaptic development. Critical for integrating new neurons into networks.	Tau inhibits PKA/CREB/BDNF signaling, impairing memory. rescue CREB phosphorylation to improve cognition.
**Dopamine (DA)**	[58]	Degeneration of VTA dopaminergic neurons reduces hippocampal DA innervation. Regulates PV interneuron activity via D2 receptors.	DA loss disrupts PV interneuron-mediated inhibition, impairing gamma oscillations.	Hippocampal overexcitation due to DA deficiency exacerbates AD pathology. L-DOPA/D2 agonists restore p-CREB levels and reduce hyperexcitability.
**Abnormally phosphorylated** **Tau proteins**	[17,59,60]	GABAergic neuronal injury, inhibiting synaptic transmission. Suppresses PKA/CREB/BDNF signaling.	Tau suppresses AHN by impairing GABAergic transmission/CREB activity.	Hyper-pTau disrupts synaptic plasticity and memory. Curcumin/melatonin reduce Tau phosphorylation and restore CREB signaling.
**cAMP/** **cGMP**	[53,54]	NO/cGMP/CREB pathway supports synaptic plasticity. Calcium/CaMKII/CREB signaling impaired by aluminum toxicity.	Enhances neurogenesis under physiological conditions.	Aβ/Tau disrupt NO/cGMP/CREB crosstalk, contributing to cognitive decline. Restoring cAMP/CREB signaling mitigates AD pathology.

**Table 2 ijms-26-06105-t002:** Neurotrophic Factors in Neurogenesis and AD Pathology.

Neurotrophic Factor	References	Mechanism in Neurogenesis	Role in AD Pathology	Therapeutic Approaches
**BDNF**	[62,63,64,65,66,68,69,74,75,76]	Promotes neural stem cell (NSC) proliferation and differentiation. Enhances synaptic plasticity and neuronal survival via TrkB signaling.	The peripheral blood BDNF level in AD patients is significantly reduced Aβ-induced TrkB-FL cleavage impairs BDNF signaling.	TAT-TrkB peptide prevents TrkB cleavage, rescues synaptic defects. Exosomes/nanocarriers deliver BDNF to the brain.
**NGF**	[65,67,70,71,77]	Supports cholinergic neuron survival and differentiation. Enhances synaptic repair.	Cholinergic neuron degeneration in AD leads to cognitive deficits.	NGF gene therapy improves cognition in models of familial AD.
**NT-3/NT-4**	[62]	Mimic BDNF functions in neurite outgrowth via PI3K/AKT and ERK pathways.	Limited direct evidence in AD; implied roles in neuroprotection.	Potential for flavonoid-based therapies.
**Lingo1**	[72,73]	Inhibits EGFR/PI3K/AKT pathway, increasing neuronal apoptosis. Promotes myelin loss.	Upregulation in AD induces Tau hyperphosphorylation via RhoA/ROCK1 signaling.	Genetic reduction of Lingo1 ameliorates cognitive dysfunction and Tau pathology.
**p75NTR**	[72,73]	Interacts with proNGF to regulate neuronal survival. Modulates AKT/GSK3β pathway.	Overactivation exacerbates Tau pathology in frontotemporal dementia (FTD) and AD.	p75NTR inhibition rescues memory deficits and reduces Tau phosphorylation.
**proBDNF**	[75]	Precursor to mature BDNF; roles in synaptic pruning.	Elevated hippocampal proBDNF correlates with pTau in AD.	Targeting proBDNF/mature BDNF balance may reduce Tau burden.

**Table 3 ijms-26-06105-t003:** Natural Bioactive Compounds in Neurogenesis and AD Pathology.

Compound/Source	References	Mechanism in Neurogenesis	Impact on AD Pathology	Key Findings
**Resveratrol**	[78,79,99]	Activates SIRT1 and Wnt/β-catenin pathways. Stabilizes β-catenin via GSK-3β phosphorylation (Ser9), inducing NeuroD1-mediated differentiation.	Enhances hippocampal neurogenesis. Improves cognitive function in models of familial AD.	Synergizes with Wnt signaling to mitigate Aβ-induced neurogenesis deficits.
**Curcumin**	[80,81]	Upregulates Notch1/Hes1, CDK4/cyclin D1, and NICD. Activates PI3K/AKT, GSK3β/Wnt, and CREB/BDNF pathways.	Reduces Aβ plaques and Tau hyperphosphorylation. Rescues cognitive deficits in APP/PS1 mice.	Poor oral bioavailability; nano-encapsulation improves solubility and efficacy.
**Coronarin D** **(*Curcuma aromatica*)**	[82,98]	Promotes astrocyte differentiation via JAK/STAT signaling. Increases GFAP expression (mRNA and protein).	Role in AD unclear; potential for glial support in neuro regeneration.	Derived from wild turmeric; targets astrocyte lineage differentiation.
**Troxerutin** **(Flavonoid)**	[83]	Enhances neuronal differentiation and axon growth. Reverses Aβ42-induced axonal inhibition.	Mitigates Aβ toxicity in neural stem cells.	Rescues Aβ-induced axonal growth defects and promotes cell migration.
**Scutellaria Flavonoids (*Scutellaria baicalensis*)**	[84]	Activates CaM-CamkIV-CREB pathway. Restores synaptic plasticity.	Ameliorates Aβ-induced learning/memory deficits.	Improves neuroplasticity in Aβ-induced AD rat models.
**DISS** **(*Polygala tenuifolia*)**	[85,97]	Enhances NSC proliferation and neuronal differentiation.	Rescues cognitive deficits and neuronal damage in APP/PS1 mice.	neurogenesis in transgenic mice.
**Ginsenoside CK** **(Ginseng)**	[86]	Stimulates hippocampal neurogenesis in young (2-month) and aged (24-month) mice.	Improves cognitive function across age groups (*p* < 0.01–0.001).	Increased hippocampal neurogenesis (5–15 mg/kg).
**Dragon’s Blood Extract** **(*QLX-N*)**	[87,90,99,100]	Induces BDNF and activates MAPK/AKT pathway. Enhances BrdU+/Nestin+ proliferation (116% and DCX+ differentiation (2.3×).	Improves spatial learning and motor function in models of familial AD.	No reported adverse effects.
**γ-CD MOFs (Curcumin)**	[91,92]	Increases curcumin solubility (53× with bile salts + γ-CD MOFs). Enhances bioavailability (16% vs. 2% without MOFs).	Facilitates sustained delivery of curcumin for AD therapy.	In vitro studies confirm improved stability and controlled release.

**Table 4 ijms-26-06105-t004:** Non-Pharmacological Interventions in Neurogenesis and AD Pathology.

Intervention Type	References	Mechanism in Neurogenesis	Impact on AD Pathology	Key Findings
**Physical Exercise**	[63,93,94,95,96,97,98]	Induces BDNF upregulation. Increases cell proliferation, migration, and dendritic spine remodeling.	Reduces neuroinflammation. Enhances cognitive function via BDNF-TrkB pathway activation.	Exercise improves neural network re-establishment in 3xTg-AD mice, delaying cognitive decline.
**Environmental Enrichment (EE)**	[68,91,99]	Boosts DCX+ neurons in DG. Activates MAPK/ERK and PI3k-AKT pathways to enhance synaptic plasticity.	Improves learning/memory in models of familial AD.	Six weeks of EE significantly benefits young mice but has limited applicability in severe AD cases.
**Ex Vivo Stem Cell Therapy**	[20,39]	Promotes synaptic integration and lesion recovery via transplanted stem cells. Uses allogeneic NSC transplantation.	Repairs neuronal loss and restores neural function.	Challenges include low cell survival and immune rejection; currently experimental.
**Combined Interventions**	[107,108,109]	Combines exercise with Mediterranean diet (antioxidants/anti-inflammatory). Nanoparticles enhance BBB penetration.	Synergistically reduces inflammation and improves cognition.	Nanoparticle delivery systems improve targeting and efficacy of neurogenic factors.

## Data Availability

Not applicable.

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
