# Peer review of "Hippocampal Neurogenesis in Alzheimer’s Disease: Multimodal Therapeutics and the Neurogenic Impairment Index Framework"

_ijms, 2025, doi:10.3390/ijms26136105_

Round 1

Reviewer 1 Report

Comments and Suggestions for Authors

This review synthesizes the knowledge on hippocampal neurogenesis in Alzheimer’s disease in detail, and introduces the Neurogenesis Impairment Index (NII) as a multidimensional framework for pre-clinical assessment and personalized intervention. The manuscript has potential to advance the field with regard to identifying effective treatments.

While the attached pdf includes point-by-point comments, there are several more general dimensions to be improved:

Wording/Grammar:

  • "AD models" is a vague and incorrect term, as ~5-10% of actual AD cases are of (mono-)genetic etiology; in contrast to the majority of sporadic AD cases where driving factors are most likely very heterogenous. Therefore, I propose to change this throughout the manuscript.
  • There are several inconstent or unclear phrases, grammatical issues and references missing. Please address them as indicated in the pdf.

Concept:

  • While acknowledging the in-depth review of Hc-AHN, the authors should also note the limits of this approach. I advise to include a more critical appraisal of how promising the approach really is, since in sporadic AD the etiology is heterogenous and multifactorial, and it is hard to believe that promoting neurogenesis in one brain area will counteract the process.
  • The actual consecutive pathophysiological sequence of where AHN fits in Abeta and Tau pathology, as well as synapse pathology is not clearly stated. The authors should provide a more consistent and elaborate concept.
  • The authors should reassess the earliest regions of AD pathology according to Braak & Braak et al. (see pdf)
  • The NII is an interesting proposal. However, its use case and their dimensions are inconsistently descirbed in the abstract and main text. The authors should clarify the number of dimensions and clearly describe how the NII can be implemented. Moreover, as it seems a central proposal of the manuscript, it is mentioned only 3 times throughout the manuscript. 

Overall, I recommend to consider the mansucript for publication after major revisions. 

Author Response

Reviewer 1

This review synthesizes the knowledge on hippocampal neurogenesis in Alzheimer’s disease in detail, and introduces the Neurogenesis Impairment Index (NII) as a multidimensional framework for pre-clinical assessment and personalized intervention. The manuscript has potential to advance the field with regard to identifying effective treatments.

While the attached pdf includes point-by-point comments, there are several more general dimensions to be improved:

Wording/Grammar:

"AD models" is a vague and incorrect term, as ~5-10% of actual AD cases are of (mono-)genetic etiology; in contrast to the majority of sporadic AD cases where driving factors are most likely very heterogenous. Therefore, I propose to change this throughout the manuscript.

Response: Thank you for this important and insightful suggestion. We fully agree that the term “AD models” may lead to conceptual ambiguity, especially given the fundamental etiological differences between familial and sporadic forms of AD. We have carefully revised the manuscript by replacing “AD models” with “models of familial AD”. These changes have been implemented throughout the manuscript, including in lines 20–21, 42, Table 2, lines 425–426, Table 3, and Table 4.

There are several inconstent or unclear phrases, grammatical issues and references missing. Please address them as indicated in the pdf.

Response: Thank you for your meticulous review and constructive feedback. We have carefully revised the manuscript to address all the issues you identified, including unclear or inconsistent expressions, grammatical inaccuracies, and missing references. Specifically, we refined the language to improve clarity and coherence, corrected grammatical errors to enhance readability, and incorporated all necessary references as indicated in your annotated PDF. All revisions have been made throughout the manuscript and are highlighted in red for ease of review. We sincerely appreciate your valuable input, which has significantly improved the quality of our manuscript.

Concept:

  1. While acknowledging the in-depth review of Hc-AHN, the authors should also note the limits of this approach. I advise to include a more critical appraisal of how promising the approach really is, since in sporadic AD the etiology is heterogenous and multifactorial, and it is hard to believe that promoting neurogenesis in one brain area will counteract the process.

Response: Thank you for your insightful and critical comments. We fully agree that AD, particularly the sporadic form, is etiologically complex and multifactorial, involving widespread neurodegenerative processes that are unlikely to be reversed by interventions targeting a single brain region.

While adult neurogenesis occurs in multiple brain regions, such as the subventricular zone and the hippocampal dentate gyrus [1], pathological changes in AD are known to originate early in the limbic system and hippocampus [2]. Therefore, this review focuses on AHN and its association with AD progression. We recognize that hippocampal neurogenesis alone cannot account for the full complexity of AD pathology. Our aim is not to overstate its therapeutic potential but to highlight hippocampal neurogenesis as one of several dimensions that may complement current pathological frameworks. We have included a more balanced and critical discussion in the revised manuscript, emphasizing that while AHN decline may contribute to disease progression, its role should be interpreted with caution and further validated in longitudinal and multimodal studies.

  1. The actual consecutive pathophysiological sequence of where AHN fits in Abeta and Tau pathology, as well as synapse pathology is not clearly stated. The authors should provide a more consistent and elaborate concept.

Response: Thank you for your insightful and important comment. We have revised the manuscript to provide a more coherent and detailed discussion of the temporal relationship between AHN impairment and the emergence of AD pathological features, including Aβ accumulation, Tau phosphorylation, and synaptic dysfunction. Specifically, we incorporated recent findings from both transgenic mouse models and human studies to demonstrate that AHN deficits often occur prior to the overt deposition of amyloid plaques and neurofibrillary tangles. These revisions can be found in lines 50–65 of the revised manuscript. We believe this addition offers a clearer and more comprehensive conceptual framework.

  1. The authors should reassess the earliest regions of AD pathology according to Braak & Braak et al. (see pdf)

Response: Thank you for this valuable suggestion. We have carefully revised the manuscript to reflect the established staging of AD pathology proposed by Braak and Braak. Specifically, we now clarify that the earliest pathological alterations in AD typically begin in the locus coeruleus and the transentorhinal cortex, and progressively affects the hippocampus. This correction has been made in lines 185–186 of the revised manuscript, with appropriate citation to Braak et al. (Acta Neuropathol. 2006 Oct;112(4):389–404. doi: 10.1007/s00401-006-0127-z).

  1. The NII is an interesting proposal. However, its use case and their dimensions are inconsistently descirbed in the abstract and main text. The authors should clarify the number of dimensions and clearly describe how the NII can be implemented. Moreover, as it seems a central proposal of the manuscript, it is mentioned only 3 times throughout the manuscript. 

Response: Thank you for your highlighting this important issue regarding the NII. We have revised both the abstract and main text to ensure consistent and precise descriptions of the NII framework.

Specifically, we now clarify that the NII is a conceptual, multidimensional index designed to characterize the degree of hippocampal neurogenesis impairment relative to core pathological and clinical features of AD. In revised lines 10–28 (abstract) and 152–164 (main text), we provide a clear definition of the components, potential scoring logic, and application scenarios.

To reflect its importance, we have also expanded the discussion of the NII throughout the manuscript, referencing it in multiple sections to emphasize its potential utility in both research and translational settings. We hope these revisions resolve the concern and strengthen the conceptual clarity and integration of the NII framework.

Overall, I recommend to consider the mansucript for publication after major revisions. 

Suggestions from manuscript PDF

Abstract

  1. not in abstract, rephrase
  2. again -> move to results
  3. What is the index meant to explain or which use cases are considered?
  4. I only read 2 dimensions hereafter: neurogeni potentieal and cognitive trajectory. please check and correct
  5. unclear abbreviations, spell them out or breifly explain
  6. again, where do these measures come from? Haven't been introduced before

Response: Thank you for your valuable comments. In response to the concerns raised, we have comprehensively restructured and refined the abstract to ensure clarity, conceptual coherence, and consistency with the format of a review article. Specifically:

We have removed any implications of original experimental results from the abstract and rephrased the content to emphasize the integrative and conceptual nature of the proposed framework.

To address concerns regarding unclear structure and dimensionality, we clarified that the NII is a novel composite metric that quantifies hippocampal neurogenic capacity relative to amyloid burden, while adjusting for demographic and cognitive variables. We explicitly stated that the NII is a conceptual index, not derived from new experimental data in this review, but instead synthesized from previously published findings to support hypothesis generation and potential translational applications.

All abbreviations have been spelled out and briefly explained at first mention to ensure accessibility for a broad readership.

We believe these revisions substantially improve the clarity and scientific relevance of the abstract and appreciate the reviewer’s input in strengthening the manuscript.

Section1: Introduction

  1. “Adult hippocampal neurogenesis (AHN)-the continuous generation and integration of new dentate gyrus neurons-plays a critical role in maintaining cognitive plasticity, particularly in learning and emotional memory consolidation.” include a reference for this statement.

Response: Thank you for your suggestion. We have added appropriate references to support this statement. Please refer to lines 40 in the revised manuscript.

  1. “However, AD is marked by a significant reduction in neurogenic capacity, likely due to impaired proliferation and survival of neural stem and progenitor cells.” Reference?

Response: Thank you for your valuable suggestion. We have added appropriate references to support this statement. Please refer to lines 42 in the revised manuscript.

  1. AD models is not quite correct for them. Most of the models, to my knowledge, are based on overexpression or introduction of a point mutation. So they introduce a strong genetic factor that mostly do not resemble the nature of sporadic (most prevalent form of) AD. I'd suggest to name it "models of familial AD" or familial amyloisis or familial tauopathy models throughout the manuscript.

Response: Thank you for this important and insightful suggestion. We fully agree that the term “AD models” may lead to conceptual ambiguity, especially given the fundamental etiological differences between familial and sporadic forms of AD. We have carefully revised the manuscript by replacing “AD models” with “models of familial AD”. These changes have been implemented throughout the manuscript, including in lines 20–21, 43, Table 2, lines 424–425, Table 3, and Table 4.

  1. “Their dendrites extend into the hippocampal molecular layer, integrating into existing hippocampal circuits. ” Ref?

Response: Thank you for your valuable suggestion. We have added appropriate references to support this statement. Please refer to lines 70 in the revised manuscript.

  1. “This dual disruption manifests as impaired neurogenic output that correlates with the severity of cognitive deficits in AD patients.” Please specify the "dual disruption"

Response: Thank you for your valuable comment. The term "dual disruption" denotes cell intrinsic NSC deficits and extrinsic niche degradation. We have made modifications to the original description. Please refer to lines 66-68 in the revised manuscript.

Thank you for your valuable comment. We have clarified that the term “dual disruption” refers to both intrinsic NSC deficits and extrinsic niche degradation. These two interrelated mechanisms collectively lead to reduced neurogenesis in the hippocampus. This clarification has been incorporated into lines 79-82 of the revised manuscript.

  1. “Salta E et al. confirmed that alterations in AHN occur in the early stages of major pathological features, such as Aβ plaques and phosphorylated tau (τ-Tau) accumulation, as well as learning and memory deficits.” this phrase is unclear. It seems as if Abeta, Tau and memory deficits would occur within a short time span. This is not the case (Abeta + perhaps PART pathology) occur years to decades before memory deficits. Please better differentiate at which time point AHN comes into play.

Response: Thank you for this important observation. We acknowledge that our original phrasing could have implied that Aβ deposition, Tau pathology, and memory deficits occur simultaneously, which is not consistent with current understanding. In the revised manuscript, we have clarified the temporal sequence by specifying that AHN impairment has been observed to occur prior to the onset of overt cognitive decline, and in some cases, even before substantial Aβ or pTau accumulation in certain models of familial AD.

To avoid conflating distinct pathological phases, we removed ambiguous phrasing and now describe the temporal dynamics more accurately, supported by recent studies. This revision appears in lines 94-97 of the revised manuscript.

  1. “And Claudia Colussi et al. found that the numbers of DCX+/BrdU+ in the hippocampus of adult AD model mice significantly decreased(DCX+/BrdU+ decline in the number about 72.7% vs WT, p<0.001,n = 4, ANOVA) ” it is unusual to inlcude the first names in such an instance.

Note, that this study used conditional Nup153 to target Hc AHN, and that this protein as candidate target has broader expression beyond Hc, which makes cell-specific applications necessary.

Response: Thank you for your helpful comment. In accordance with academic writing conventions, we have removed the first name and now refer to the study as “Colussi et al.” in the revised manuscript.

We also appreciate your point regarding the broader expression of Nup153 beyond the hippocampus. In the revised text, we now note that although the study employed a conditional approach to investigate Nup153 function in hippocampal NSC, the protein is expressed in multiple brain regions. Therefore, any therapeutic strategy targeting Nup153 would require careful consideration of cell specificity to minimize potential off-target effects. These revisions have been incorporated into lines 101-107 of the revised manuscript.

  1. “Therapeutic targeting of AHN, Zheng J et al. reported that the tau phosphorylation accumulation in AD leads to damage in dentate gyrus GABAergic interneurons, resulting in decreased neurogenic function. Restoring the number of GABAergic neurons could im-prove AHN and enhance cognition in both AD patients and animal models[17] (δ subunit-containing γ-aminobutyric acid type A receptors(δ-GABAAR)agonist THIP (through modulating tonic inhibition via extrasynaptic δ-GABAAR receptors[18])).”Please revise this senctence for grammatical correctness. Specify "damage", synaptic dysfunction, cell loss etc? Please rephrase, as it is quite hard to read this sentence

Response: Thank you for the helpful comments. We agree that the original sentence lacked clarity and grammatical precision. In the revised manuscript, we have rewritten this section to specify the nature of the neuronal damage and to improve readability. Specifically, we now state that Tau phosphorylation in AD impairs GABAergic interneurons in the dentate gyrus, leading to reduced synaptic signaling and a consequent decline in AHN. We further clarify that restoring the number of GABAergic neurons—such as through pharmacological modulation of extrasynaptic δ-subunit-containing GABA_A receptors (δ-GABA_ARs) using the agonist THIP—can rescue neurogenic activity and improve cognitive outcomes in models of familial AD. These changes have been incorporated in lines 119-124 of the revised manuscript.

  1. “Fur-thermore, a single systemic injection of wild-type hematopoietic stem/progenitor cells into 5xFAD mice effectively prevents or alleviates various signs and symptoms of AD” which symptoms, and are these symptoms of AD? I'd rather suggest to write the description of the behavioral test used in this study.

Response: Thank you for this important comment. We have revised the sentence to specify the phenotypic outcomes observed in the study. We now state that the injection of wild-type hematopoietic stem/progenitor cells into 5xFAD mice resulted in the preservation of neurocognitive performance in novel object recognition tasks. The intervention also partially preserved blood–brain barrier integrity and reduced pathological markers such as Aβ plaque load, microgliosis, and neuroinflammation. These revisions clarify that the term “symptoms” refers to both cognitive deficits and AD-related neuropathological features. Please refer to lines 125-128 in the revised manuscript.

  1. “neural stem cells can protect against behavioral and molecular pathology in AD by signifi-cantly inhibiting Aβ plaques aggregation” please substitue with "amylosis and neuroinflammation"

Response: Thank you for the correction. We have revised the sentence accordingly in the manuscript. Please refer to lines 130 in the revised manuscript.

  1. “The immature dentate granule cells (imGC) generated by AHN contribute to the plas-ticity and unique brain function of rodents, and dysregulation of imGC is observed in various human neurological diseases. Zhou Y et al. determined that imGCs persist in the human hippocampus throughout life, with their proportion gradually decreasing with age.” ref?

Response: Thank you for your comment. We have added appropriate references to support these statements. Please refer to lines 134 -135 in the revised manuscript.

  1. “Overall consideration, we proposed the AD NII to expand the horizons for research and treatment, by comprehensively considering differences in age, gender, and cognitive impairment assessment indices of AD patients or model animals, while additionally in-corporating differences in hippocampal neurogenesis rates into the evaluation frame-work.” write out here again. As most of the studies are conducted in rodents, specificially in mice, it would be most adequate to limit it to these species, unless further tested, ior at least include a limitation statement about it.

Response: Thank you for your constructive comments. We have rephrased the description of the NII framework to clearly reflect its current basis in preclinical models. Moreover, we have added a limitation statement noting that the translational applicability of the NII to human AD patients requires further validation. These revisions have been made in lines 152-166 of the revised manuscript.

  1. “Annotations: Doublecortin (DCX) cells, which a widely accepted marker of newly generated granule cells” "which IS a"

Response: Thank you for your comment. This has been revised.

  1. Linked to the comment above: where exactly between abeta, tau pathology and cognitive impairment does AHN decline come into play?

Response: Thank you for this important follow-up question. Based on current evidence from both transgenic mouse models and human postmortem studies, AHN decline appears to occur before the initial molecular pathology (such as Aβ or tau pathology) and the onset of overt cognitive impairment.

Mechanistically, AHN dysfunction reflects an insufficient compensatory response to early neuronal and synaptic loss. In the context of AD, both intrinsic factors (e.g., impaired neural stem/progenitor cell proliferation, aberrant differentiation) and extrinsic niche disruptions (e.g., inflammation, altered signaling) contribute to reduced neurogenesis. This impairs the hippocampal network’s ability to maintain plasticity and functional homeostasis, potentially accelerating cognitive decline.

We have added a more explicit explanation of this timing and mechanism in the revised manuscript (see lines 177–182), to clarify that AHN impairment has been observed to occur prior to the onset of overt cognitive decline, and in some cases, even before substantial Aβ or pTau accumulation in certain models of familial AD.

  1. keep "Tau", "pTAU" or "tau" consistently with regard to upper/lower case letters

Response: Thank you for your comment. These have been revised as “Tau”.

Section2: Hippocampal Neural Stem Cells and AD Pathologic Processes

  1. This is not entirely correct. Please revise accoridng to Braak & Braak, see Locus coreuleus and transentorhinal cortex.

Response: Thank you for your comment. We have corrected this as “In AD, pathology begins in the transentorhinal cortex and locus coeruleus and progressively affects the hippocampus, resulting in its hallmark atrophy”.

  1. “In healthy DG, NSCs maintain differentiation equilibrium, Neural stem cells located in the SGZ of the DG in normal adult mammals can self-renew, proliferate, migrate, and differentiate” lower case

Response: Thank you for your suggestion. It has been revised.

  1. “Transplanting neural stem cells into early-stage AD rats repairs dam-aged synapses and enhances learning and memory” where?

Response: Thank you for your comment. We are sorry for this mistake. It has been revised as “AD mouse model”.

  1. “This integration will establish a novel neurogenic therapeutic index framework, thereby providing transformative perspectives for both mechanistic elucidation and clinical management strategies in AD and other neurodegenerative disorders.” as preclinical treatment trial readout?

Response: Thank you for this valuable comment. We agree that the proposed NII currently serves best as a conceptual and quantitative framework for preclinical studies, particularly as a potential readout for evaluating neurogenesis-targeted interventions in animal models. In response, we have revised the statement to clarify that the NII is designed primarily for preclinical research and mechanistic studies, and that its future clinical translation will require further validation. Please refer to lines 237-243 in the revised manuscript.

  1. “Incorporate the following advancements across AD research and therapeutic development: Establish a neurogenesis index as a quantitative biomarker for AD pathological staging and diagnostic evaluation; Implement neurogene-sis-modulating strategies to enhance cognitive performance and emotional regulation in both clinical interventions and animal model therapeutics; Prioritize neurogenesis-tar-geted drug design by screening compounds against key molecular checkpoints (e.g., BDNF/TrkB signaling, Wnt/β-catenin pathway, and Notch1 receptor dynamics).”

Is this a sub-heading? And what are the following sentences meant to convey? It is unclear why this is written in an imperative here. Please revise for clarity.

Response: Thank you for pointing this out. We agree that the original formulation was ambiguous in structure and tone. We have rephrased this section to clarify that these are proposed directions for future research and therapeutic development, not directives. The language has been revised from imperative to objective, and the section has been fully integrated into the narrative flow of the discussion. Please refer to lines 245-248 in the revised manuscript.

Section3: Research on Adult Neurogenesis in Alzheimer's Disease

  1. “The hippocampus is the first brain region to be affected in AD, and hippocampal atrophy is one of the characteristic pathologies of AD” Again, see above comment

Response: Thank you for your observation. To avoid redundancy with earlier statements regarding the regional progression of AD pathology, we have removed this sentence from the revised manuscript.

  1. “In 2022, researchers Zhou Y et al. used single-cell sequencing to discover that imma-ture granule cells (imGCs) exist in the hippocampus throughout the entire lifespan.” what kind of sequencing? RNA? ATAC?

Response: Thank you for your comment. We have revised the manuscript to specify that the study by Zhou Y et al. employed snRNA-seq to identify immature granule cells (imGCs) in the human hippocampus across the lifespan. This clarification has been added in line 260 of the revised manuscript.

Section 4: Neurotransmitters and signaling molecules Modulation of Hippocampal Neurogenesis Alleviates Alzheimer's Disease Pathology

  1. “The earliest pathological hallmarks of AD are linked to the dysfunction and degeneration of cholinergic neurons in the basal forebrain and their cortical projections.” Again, see above and mention Locus coeruleus pathology

Response: Thank you for your important observation. We have updated the description of early AD pathology to include the involvement of the locus coeruleus, in accordance with Braak staging and subsequent neuropathological studies. The sentence now reflects that the earliest pathological changes occur in the locus coeruleus and transentorhinal cortex, preceding basal forebrain cholinergic degeneration. Please refer to lines 294-295 in the revised manuscript.

  1. “The cognitive decline observed in AD is significantly influenced by the accumulation of phos-phorylated Tau proteins, which impair AHN.” USe "pTau" and provide references for this particular link.

Response: Thank you for your suggestion. It has been revised.

  1. “the ven-tral hippocampal CA1 (vCA1) region gradually accumulates with tau, vCA1,” Grammar, full stop.

Response: Thank you for your suggestion. It has been revised.

  1. “Especially its excitability and small white protein (PV) neurons are completely filled with mislocated and phosphorylated tau.” it is usually spelled "parvalbumin". please use more professional wording.

Response: Thank you for your suggestion. We have replaced it with the correct term “parvalbumin interneurons” and rephrased the sentence to ensure clarity and professional scientific tone. Please refer to lines 311-313 in the revised manuscript.

  1. “Notably, the anti-AD drug donepezil has been shown to exert neuroprotective effects by upregulating CREB…” you can't state that. You can use reversible AchE inhibitor or similar

Response: Thank you for your suggestion. It has been revised.

  1. “Changes in the function of small albumin interneurons (PV INs) were observed in Alzheimer's disease (AD)....” wording see above

Response: Thank you for your suggestion. It has been revised.

  1. “Elena Spoleti et al. used a Tg2576 AD mouse model to…” omit frist names

Response: Thank you for your suggestion. It has been revised.

  1. Table1 “Accumulates in GABAergic interneurons,”not exclusively ! Tau is neither a ntm nor signaling molecule in the narrower sense

Response: Thank you for the correction. We have revised the corresponding entry in Table 1 to replace “Tau protien” with “abnormally phosphorylated Tau proteins” and “Accumulates in GABAergic interneurons” with “GABAergic neuronal injury”.

Section6: Natural Product Active Ingredients Promoting Neurogenesis to Improve AD Pathology

  1. “Numerous studies have demonstrated that traditional Chinese herbs and their natural active components…” Is that exclusive to Chinese herbs? please change to natural compounds, some of which have been used historically.

Response: Thank you for this insightful suggestion. We have rephrased the sentence to refer more broadly to “natural compounds”, which better reflects the diversity of sources and historical context. Please refer to line 423 in the revised manuscript.

  1. “…DCX+ neuronal differentiation (2.3-fold elevation, p<0.001) in SGZ, significantly enhancing spatial learning memory and motor abilities in rat models. ” any reference to add? Otherwise add (unpublished)

Response: Thank you for your suggestion. We have added the appropriate reference to support the reported findings. Please refer to line 454 in the revised manuscript.

  1. “Notably, while promoting endogenous neurogenesis, these natural bioactive compounds have not been reported to induce excessive neurogenesis-related events such as epilepsy or carcinogenesis, indicating their pro-proliferative regulation operates through controlled mechanisms.” I think this is too far-fetched. Provide supporting evidence or state, that it is unknown, and epidemiological or direct evidence is needed.

Response: Thank you for your critical comment. We have rephrased the sentence to reflect the current state of knowledge more accurately. Specifically, we now acknowledge that while no direct evidence has linked these compounds to adverse outcomes such as epilepsy or tumorigenesis, systematic preclinical and epidemiological studies are still required to fully evaluate their long-term safety. Please refer to line 454-459 in the revised manuscript.

Section7: Non-Pharmacological Interventions and Future Directions

  1. “Cognitive dysfunction represents a significant clinical challenge in AD, progressively worsening as the disease advances…” this can be easily summarized as: "Progressive cognitive decline in AD".

Response: Thank you for your suggestion.  It has been revised as “Progressive cognitive decline in AD”.

  1. “as they are not suitable for individuals with severe cognitive impairment or compromised motor function.” not necessarily suitable

Response: Thank you for your suggestion.  It has been revised as “not necessarily suitable”.

  1. “In humans, the DG produces approximately 700 newborn neurons daily(C14 concen-tration in genomic DNA of hippocampal neurons (n=55) and non neuronal cells (n=65)), allowing for the replacement of about 35% of DG neurons rover a lifetime, and rodents replace about 10%. ” rephrase for improved clarity

Response: Thank you for your suggestion. In response, we have revised the sentence to improve clarity and readability. The updated version now reads: "Human DG neurogenesis is estimated at ~700 new neurons per day, resulting in ~35% neuronal turnover over the lifespan, as compared to ~10% in rodents." This revised sentence more concisely conveys the key quantitative comparison and has been incorporated in line 517-518 of the revised manuscript.

  1. “Increasing evidence indicates that pro-inflammatory cytokines and inflamma-tory molecules(Under neuroinflammatory conditions, IL-1β and TNF-α signaling through the NF-κB pathway leads to impaired hippocampal) released during neuroinflammation negatively modulate hippocampal neurogenesis, particularly in the context of AD, dis-rupting the dynamics of the neurogenic niche(including diminished BDNF release from cerebrovascular endothelial cells, which is known to support neuronal survival and dif-ferentiation) . ” rephrase for clarity, try to avoid parenthesis

Response: Thank you for the helpful suggestion. To improve clarity and readability, we have rephrased the sentence by converting parenthetical content into full clauses. The revised version now reads: "Increasing evidence indicates that pro-inflammatory cytokines and inflammatory mediators released during neuroinflammation negatively regulate hippocampal neurogenesis, particularly in the context of Alzheimer’s disease. For example, IL-1β and TNF-α signaling through the NF-κB pathway has been shown to impair hippocampal neurogenesis by disrupting neural stem cell function. In addition, neuroinflammation can reduce the release of brain-derived neurotrophic factor (BDNF) from cerebrovascular endothelial cells, which normally supports neuronal survival and differentiation, thereby further compromising the neurogenic niche." This revision appears in lines 527-531 of the revised manuscript.

  1. “Thus, regulating the developmental homeostasis of hippocampal neural stem cells could represent a key break-through in elucidating AD’s pathological mechanisms and in developing effective diagnosis and treatment strategies.” this is quite optimisitc. Please also include inherent limitaitons of Hc-focused or AHN-directed therapeutic approaches. this includes, that timing might be too late in many patients, areas other than the Hc are also involved an pathology-spread can bypass the Hc route. Furthermore, co-pathology - which is very frequent - will co-occur and may negatively impact the resutls of AHN improvement.

Response: Thank you for this insightful and balanced comment. We fully agree that while targeting hippocampal neurogenesis holds therapeutic potential, several important limitations must be considered. In the revised manuscript, we have modified the original statement to temper its tone and included a discussion of the key limitations of AHN-directed strategies. These include the risk of delayed therapeutic timing, the involvement of multiple brain regions in AD pathology beyond the hippocampus, and the frequent presence of co-pathologies that may compromise the efficacy of neurogenesis-based interventions. This revision has been incorporated into lines 543-555 of the revised manuscript.

References:

  1. Bond, A. M., G. L. Ming, and H. Song. "Adult Mammalian Neural Stem Cells and Neurogenesis: Five Decades Later." Cell Stem Cell 17, no. 4 (2015): 385-95.
  2. Salta, E., O. Lazarov, C. P. Fitzsimons, R. Tanzi, P. J. Lucassen, and S. H. Choi. "Adult Hippocampal Neurogenesis in Alzheimer's Disease: A Roadmap to Clinical Relevance." Cell Stem Cell 30, no. 2 (2023): 120-36.

Reviewer 2 Report

Comments and Suggestions for Authors

Hippocampal neurogenesis impairment is referred as a pathological hallmark of Alzheimers`s disease. In this review, the authors described different kind of therapeutical strategies targeting hippocampal neurogenesis, such as neurotransmitters and signaling molecules, neurotrophic factors, bioactive compounds and non-pharmacological strategies. The manuscript is clear and well organized, however I have some suggestions to improve it:

  1. Please add one reference to the formula related to the calculation of neurogenic capacity.
  2. Paragraph 3. "Currently, some studies have applied stem cells from the placenta, adipose tissue, umbilical cord blood, and bone marrow to the research and treatment of Alzheimer`s disease." Please add references. 
  3. Paragraph 4. "Similarly, choline levels in the basal forebrain; The expression of hippocampal N-acetyl aspartate, basal forebrain ChAT and vesicular acetylcholine transporter (vAchT); the Muscarinic acetylcholine receptor 2 (CHRM2) in the hippocampus are increased." I suggest to adjust the punctuation of this period.
  4. In paragraph 6 the authors talk about bioactive compounds having a role in promoting neurogenesis. However, in addition to those mentioned in the text, there are other bioactive compounds whose neurogenic potential has been proved. I talk about lactoferrin which is proved to improve neurogenesis in the hippocampus (DOI: 10.1007/s10517-024-06004-3) and promote the reprogramming of the astrocytes in neural precursor cells (DOI: 10.3390/ijms26010405). Other examples are catalpol (DOI: 10.3389/fphar.2024.1461279), salidroside (DOI: 10.1017/neu.2024.28) and rosmarinic acid (DOI: 10.1016/j.biopha.2023.115994). Please, add these references to the text.

Author Response

Reviewer 2

Hippocampal neurogenesis impairment is referred as a pathological hallmark of Alzheimers`s disease. In this review, the authors described different kind of therapeutical strategies targeting hippocampal neurogenesis, such as neurotransmitters and signaling molecules, neurotrophic factors, bioactive compounds and non-pharmacological strategies. The manuscript is clear and well organized, however I have some suggestions to improve it:

  1. Please add one reference to the formula related to the calculation of neurogenic capacity.

Response: Thank you for your expert suggestion. We have added relevant references to support the conceptual basis and rationale behind the formula used to estimate neurogenic capacity. The references appear adjacent to the formula description in lines 162 of the revised manuscript.

  1. Paragraph 3. "Currently, some studies have applied stem cells from the placenta, adipose tissue, umbilical cord blood, and bone marrow to the research and treatment of Alzheimer`s disease." Please add references. 

Response: Thank you for your suggestion. We have added representative references supporting the application of stem cells derived from placenta, adipose tissue, umbilical cord blood, and bone marrow in AD research and therapeutic exploration. These studies provide evidence for the neuroprotective and regenerative potential of various stem cell sources in models of familial AD. The relevant references have been incorporated into the revised manuscript as citations [64–68]. These additions appear in paragraph 3, lines 279 of the revised manuscript.

  1. Paragraph 4. "Similarly, choline levels in the basal forebrain; The expression of hippocampal N-acetyl aspartate, basal forebrain ChAT and vesicular acetylcholine transporter (vAchT); the Muscarinic acetylcholine receptor 2 (CHRM2) in the hippocampus are increased." I suggest to adjust the punctuation of this period.

Response: Thank you for your suggestion. We have revised the sentence for improved punctuation, clarity, and grammatical correctness. This revision has been incorporated into lines 300-302 of the revised manuscript.

  1. In paragraph 6 the authors talk about bioactive compounds having a role in promoting neurogenesis. However, in addition to those mentioned in the text, there are other bioactive compounds whose neurogenic potential has been proved. I talk about lactoferrin which is proved to improve neurogenesis in the hippocampus (DOI: 10.1007/s10517-024-06004-3) and promote the reprogramming of the astrocytes in neural precursor cells (DOI: 10.3390/ijms26010405). Other examples are catalpol (DOI: 10.3389/fphar.2024.1461279), salidroside (DOI: 10.1017/neu.2024.28) and rosmarinic acid (DOI: 10.1016/j.biopha.2023.115994). Please, add these references to the text.

Response: Thank you for your insightful suggestions and valuable references. We have added citations for lactoferrin, catalpol, salidroside, and rosmarinic acid in the revised text to reflect recent advances in bioactive compounds that promote adult hippocampal neurogenesis. The sentence now reads: “Furthermore, compounds including lactoferrin, catalpol, salidroside, and rosmarinic acid have been shown to regulate neurogenesis and related pathological processes.”

We have also updated Table 3 to include the categorized regulatory mechanisms of these compounds on neural stem cell proliferation, differentiation, and neurogenic reprogramming. These additions are reflected in the revised manuscript (lines 469-471) and corresponding references [118–120].

Round 2

Reviewer 1 Report

Comments and Suggestions for Authors

Thank you for your thorough and thoughtful revisions to the manuscript. The extensive changes have addressed the majority of previous concerns, and I acknowledge the significant effort invested by the authors.

Minor Comments:

  • Page 2: 

    • Please correct "subject of debate debated."

    • Note that "MCI" is not necessarily an AD prodrome and can precede various forms of dementia.

    • The authros should ensure that gene names are italicized and protein names are in bold, following international conventions and specifying murine or human context as appropriate.

  • Page 3: Please spell out "THIP" at first mention, or omit if not essential.

  • Page 4: The NII is the central proposal of this review. However, the translation of the 3rd NII dimension into clinical application remains unclear. Please elaborate on how DCX cell/SGZ volume quantification could be approached in living patients, or clarify if this particular metric is only relevant in preclinical animal models. If so, specify how cognitive assessments such as MoCA would fit in these preclinical models. This section would benefit from a clearer discussion of when and in which species context each dimension should be studied.

  • Page 8: Please correct "PV neuron disfunction dysfunction" 

Recommendation:
Accept with minor revisions.

Author Response

Dear Editors and Reviewers,

We are pleased to have received your valuable comments and suggestions on our manuscript titled "Hippocampal Neurogenesis in Alzheimer’s Disease: Multimodal Therapeutics and the Neurogenic Impairment Index Framework" (Manuscript ID: ijms-3630497). We have carefully considered each of your points and revised the manuscript accordingly. In the revised manuscript, all changes made to the text and figures are highlighted in red.

Attached, you will find detailed point-by-point responses to the reviewers’ comments. We hope these revisions address your concerns and meet the publication standards of the International Journal of Molecular Sciences. If further revisions are required, please do not hesitate to let us know. Thank you for your time and consideration.

Sincerely,

Liang Yang

Research Center for Natural Peptide Drugs, Shaanxi Engineering & Technological Research Centre for Conservation & Utilization of Regional Biological Resources, Yanan University, Yanan 716000, China.

Email: lyang@yau.edu.cn

Thank you for your thorough and thoughtful revisions to the manuscript. The extensive changes have addressed the majority of previous concerns, and I acknowledge the significant effort invested by the authors.

Minor Comments:

Page 2:

Please correct "subject of debate debated."

Thank you for the reviewer's highly professional suggestion to change "remains a subject of debated" to "subjects of ongoing debate" in the manuscript. See line 50.

Note that "MCI" is not necessarily an AD prodrome and can precede various forms of dementia.

We thank the reviewers for their highly professional suggestion. According to the original description in the cited literature—"we demonstrate that adult hippocampal neurogenesis is present from the eighth to the tenth decade of life and that it is detectable even in persons with mild cognitive impairments (MCI) and AD" —MCI and AD are treated as distinct conditions. Therefore, we have removed the parenthetical phrase "—a prodromal stage of AD—" from the manuscript (see line 58).

The authros should ensure that gene names are italicized and protein names are in bold, following international conventions and specifying murine or human context as appropriate.

We thank the reviewers for their highly professional suggestions. All recommended revisions have been addressed throughout the manuscript. Due to the extensive nature of these changes, we have not enumerated them individually here.

Page 3: Please spell out "THIP" at first mention, or omit if not essential.

We thank the reviewers for their professional and meticulous suggestions. Per their recommendation, we have added the full chemical name "4,5,6,7-tetrahydroisoxazolo[5,4-c]pyridin-3-ol" before the abbreviation "THIP" in the manuscript (see line 128).

Page 4: The NII is the central proposal of this review. However, the translation of the 3rd NII dimension into clinical application remains unclear. Please elaborate on how DCX cell/SGZ volume quantification could be approached in living patients, or clarify if this particular metric is only relevant in preclinical animal models. If so, specify how cognitive assessments such as MoCA would fit in these preclinical models. This section would benefit from a clearer discussion of when and in which species context each dimension should be studied.

We thank the reviewer for highlighting this key translational issue. We agree that the third dimension of the NII—namely, “neurogenic capacity,” which includes quantification of DCX+ cells and SGZ volume—is currently limited to preclinical studies due to the invasive nature of tissue-based neurogenesis markers.

To address this, we have revised the manuscript (lines 161-187) to clarify that the structural component of the NII is specifically designed for use in animal models, particularly transgenic mouse models of AD, where histological analyses can be performed. In contrast, human applications of the NII would rely on non-invasive proxies, such as CSF biomarkers (Aβ42/p-tau), neuroimaging readouts, or emerging peripheral blood indicators, alongside cognitive assessments such as the Montreal Cognitive Assessment (MoCA). We also added discussion of how rodent cognitive assessments (e.g., Morris Water Maze) can serve as translational equivalents to human cognitive tools like MoCA, and how performance is evaluated relative to normative age-matched controls.

These clarifications help delineate the scope and species-specific application of each NII domain, thereby enhancing the translational relevance and practical implementation of the index.

Page 8: Please correct "PV neuron disfunction dysfunction"

Thank you for catching this oversight. We have corrected “PV neuron disfunction dysfunction” to “dysfunction of PV interneurons” to align with standard terminology and the cited literature (line 366).
